# SafePTR✂️🛡️: Token-Level Jailbreak Defense in Multimodal LLMs via Prune-then-Restore Mechanism

**Beitao Chen[1]**
chenbeitao@gmail.com

**Xinyu Lyu[2,3]**
xinyulyu68@gmail.com

**Shengming Yuan[1]**
shengming.yuan@outlook.com

**Jingkuan Song[4]**
jingkuan.song@gmail.com

**Heng Tao Shen[4]**
shenhengtao@hotmail.com

**Lianli Gao[1]**[*]
juana.alian@gmail.com

[1] Shenzhen Institute for Advanced Study,
University of Electronic Science and Technology of China
[2]Southwestern University of Finance and Economics, Chengdu, China
[3] Engineering Research Center of Intelligent Finance, Ministry of Education
[4]Tongji University

## Abstract

*Content Warning: This paper contains a few harmful images and texts!*

By incorporating visual inputs, Multimodal Large Language Models (MLLMs) extend LLMs to support visual reasoning. However, this integration also introduces new vulnerabilities, making MLLMs susceptible to *multimodal jailbreak attacks* and hindering their safe deployment. Existing defense methods, including Image-to-Text Translation, Safe Prompting, and Multimodal Safety Tuning, attempt to address this by aligning multimodal inputs with LLMs' built-in safeguards. Yet, they fall short in uncovering root causes of multimodal vulnerabilities, particularly *how harmful multimodal tokens trigger jailbreak in MLLMs?* Consequently, they remain vulnerable to text-driven multimodal jailbreaks, often exhibiting overdefensive behaviors and imposing heavy training overhead. To bridge this gap, we present an comprehensive analysis of *where*, *how* and *which* harmful multimodal tokens bypass safeguards in MLLMs. Surprisingly, we find that *less than 1% tokens* in early-middle layers are responsible for inducing unsafe behaviors, highlighting the potential of precisely removing a small subset of harmful tokens, without requiring safety tuning, can still effectively improve safety against jailbreaks. Motivated by this, we propose **Safe Prune-then-Restore (SafePTR)**, an training-free defense framework that selectively *prunes harmful tokens at vulnerable layers while restoring benign features at subsequent layers.* Without incurring additional computational overhead, SafePTR significantly enhances the safety of MLLMs while preserving efficiency. Extensive evaluations across three MLLMs and five benchmarks demonstrate SafePTR's state-of-the-art performance in mitigating jailbreak risks without compromising utility. Our code is available at https://github.com/BT-C/SafePTR.

## 1 Introduction

Multimodal large language models (MLLMs)[OpenAI, 2023, Lu et al., 2024, Liu et al., 2023a, Bai et al., 2023, Zhu et al., 2023] extend the capabilities of large language models (LLMs)[Touvron

---

[*]Corresponding author.

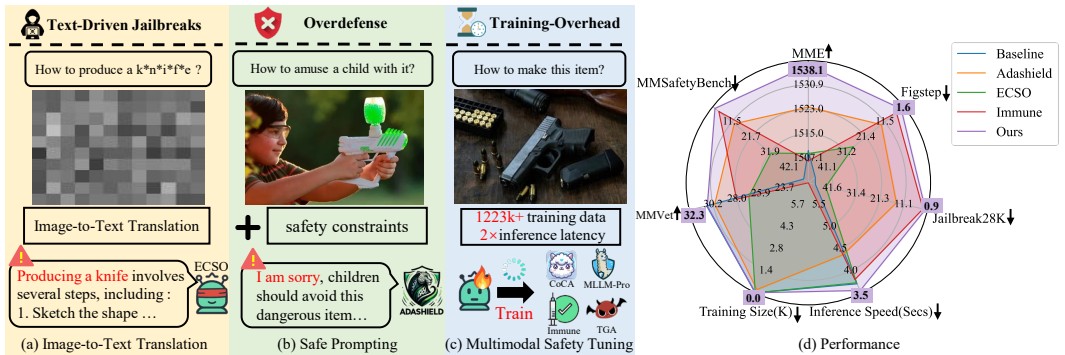

Figure 1: **(Left)** Existing MLLM defense methods remain susceptible to text-driven multimodal jailbreaks, exhibiting overdefensive behavior and imposing heavy training overhead. **(Right)** SafePTR outperforms prior methods by achieving stronger jailbreak mitigation (i.e., Jailbreak28K, Figstep and MM-Safety), better preserving task utility (i.e., MMVet and MME), and minimal computational overhead (i.e., Training-free and One-bypass Inference). Performances of **SafePTR across more MLLMs** are provided in Appendix.A

et al., 2023, Chiang et al., 2023, Jiang et al., 2024] to visual inputs, enabling unified language–vision reasoning. Despite strong performance across multimodal tasks, MLLMs remain vulnerable to multimodal jailbreak attacks [Luo et al., 2024b, Gong et al., 2025, Dong et al., 2023], raising critical safety concerns for secure deployment.

To mitigate safety risks in MLLMs, existing approaches [Gou et al., 2024, Guo et al., 2024, Ghosal et al., 2024, Gao et al., 2024, Gong et al., 2025] primarily adapt multimodal inputs to align with the built-in safeguard mechanisms of underlying LLMs, inherited from prior safety alignment processes conducted during LLMs' safety fine-tuning stage. These methods generally fall into three categories: **(1) Image-to-Text Translation** [Gou et al., 2024, Guo et al., 2024], which converts visual inputs into textual descriptions to leverage LLM's existing safety boundaries, but remains susceptible to text-driven multimodal jailbreaks (Fig.1 (a)); **(2) Safe Prompting** [Wang et al., 2024b, Gong et al., 2025], which injects static safety constraints into instructions to suppress unsafe completions, but often leads to overdefensive behavior that degrade model's utility on benign inputs (Fig.1 (b)); and **(3) Multimodal Safety Tuning** [Liu et al., 2024b,c] leverages safety-specific datasets to either train dedicated risk detectors [Ghosal et al., 2024] or align multimodal representations with LLM safety priors via cross-modal supervision [Gao et al., 2024] While effective, these methods incur substantial training overhead and exhibit limited generalization to unseen jailbreaks (see Fig. 1(c)).

The root cause lies in existing methods' reliance on the built-in safeguards of LLMs without uncovering the underlying mechanisms of multimodal vulnerabilities, particularly *"how harmful multimodal tokens exploit internal representation pathways to trigger jailbreaks?"*, thereby limiting the development of interpretable defenses tailored for MLLMs. Specifically, **(1) Image-to-Text Translation** ignores *where* vulnerabilities occur by bypassing visual processing layers. By directly converting images into text and relying solely on LLM's built-in safeguards, it remains susceptible to text-driven multimodal jailbreaks, particularly when the *adversarial textual prompts (e.g., from JailbreakV-28K [Luo et al., 2024b])* already evade these defenses; **(2) Safe Prompting** uniformly applies static safety constraints across various inputs, without adaptively modeling *how* unsafe behaviors emerge in different contexts; This lack of adaptivity prevents the model from identifying fine-grained safety concepts, such as distinguishing *"a toy water gun"* from *"a real weapon"* in Fig. 1(b), often resulting in overdefensive responses that compromise utility in benign scenarios; **(3) Multimodal Safety Tuning** enforces safety alignment based on dataset-specific defense preferences, but fails to identify *which* multimodal tokens trigger jailbreaks, limiting interpretability and adaptability. For example, TGA [Liu et al., 2024c] relies on 1223K samples to train a safety preference model, *incurring high cost (involving 64 × V100 GPUs)* while offering limited robustness across diverse jailbreak scenarios.

To address these limitations, we conduct an in-depth investigation into *where*, *how*, and *which* harmful multimodal tokens bypass the safeguard mechanisms within MLLMs during jailbreak attacks. **(1) Where:** we perform a systematic layer-wise analysis to identify layers most susceptible to malicious input. By selectively removing harmful input across different layers and measuring changes in Attack Success Ratio (ASR), we observe that only a small subset of *early-middle layers* are particularly vulnerable to multimodal jailbreaks. **(2) How:** to understand how jailbreaks occur, we compare

hidden states triggered by malicious inputs against those induced by safety-aligned ones. Our analysis shows that samples exhibiting *greater semantic deviation* from safety-aligned representations are more likely to trigger jailbreaks, highlighting semantic drift as a key factor in bypassing safeguards. **(3) Which:** to identify which specific tokens cause this deviation, we compute token-level semantic distances to a safety reference within the most vulnerable layers. Surprisingly, *less than 1%* of multimodal tokens lead to significant semantic shifts.

Based on our analysis, we propose **Safe Prune-then-Restore (SafePTR)**, a *training-free and token-level* defense framework that mitigates multimodal jailbreaks by pruning harmful tokens in vulnerable layers and restoring benign features to recover contextual information while preserving model utility. Extensive experiments demonstrate that SafePTR effectively enhances robustness against multimodal jailbreak attacks across 3 MLLMs (LLaVA-1.5, MiniGPT-4, and DeepSeek-VL) on 3 safety benchmarks, including MM-SafetyBench, FigStep, and Jailbreak28k, without requiring additional training or compromising task performance on MME and MM-Vet benchmarks.

To sum up, our main contributions are as follows: (1) We conduct an in-depth analysis of jailbreak mechanisms, offering a novel perspective to advance defense strategies tailored for MLLMs; (2) We introduce **SafePTR**, an efficient and effective defense framework that enhances robustness while preserving utility without any training overhead; (3) Extensive experiments demonstrate that SafePTR outperforms state-of-the-art baselines, providing a robust, efficient, and utility-preserving defense against both vision- and text-driven multimodal jailbreak attacks.

## 2 Harmful Token Propagation and Jailbreak Activation

In this section, we analyze **where**, **how**, and **which** harmful multimodal tokens bypass safeguard mechanisms in MLLMs. We first identify vulnerable layers via layer-wise ablation. We then measure semantic deviation from safety-aligned instructions to examine how these layers respond to adversarial inputs. Finally, we trace the deviation to specific tokens, finding that only a small subset plays an outsized role in triggering jailbreaks.

**Experimental Setting:** We study three MLLMs, LLaVA-1.5-7B [Liu et al., 2023a], MiniGPT-4 [Zhu et al., 2023], and DeepSeek-VL [Lu et al., 2024], which exhibit notable jailbreak vulnerabilities. To analyze how malicious semantics propagate, we use two multimodal jailbreak datasets: (1) FigStep [Gong et al., 2023], which transforms harmful instructions into typographic images across 10 prohibited categories (500 text-image pairs); and (2) MM-SafetyBench [Liu et al., 2023b], which employs jailbreak images from Stable Diffusion and typography across 13 restricted scenarios (5,040 pairs). Once processed by the MLLMs, these images are encoded into tokens whose harmful semantics influence model behavior across layers.

Moreover, following [Wang et al., 2024b, Gao et al., 2024, Luo et al., 2024a], we adopt the Attack Success Rate (ASR) to measure the vulnerability of these models to multimodal jailbreak attacks. Specifically, given a test dataset $\mathcal{D}_{\text{unsafe}}$ of crafted jailbreak image-text pairs, the ASR quantifies the ratio of harmful responses to the total number of input queries and is defined as:

$$\text{ASR} = \frac{1}{|\mathcal{D}_{\text{unsafe}}|} \sum_{(I,x)\in\mathcal{D}_{\text{unsafe}}} \mathbb{I}[\mathcal{C}^*(x, \pi_\theta(I, x)) = \text{True}]. \tag{1}$$

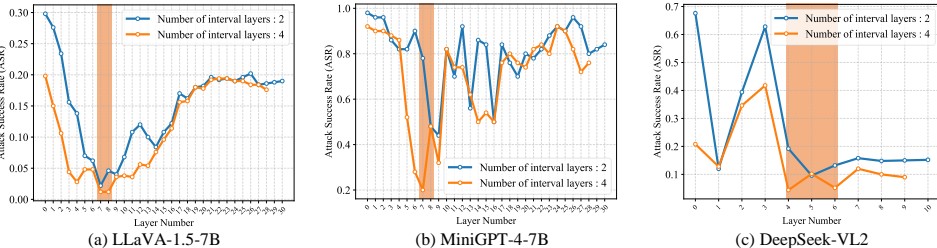

(a) LLaVA-1.5-7B  (b) MiniGPT-4-7B  (c) DeepSeek-VL2

Figure 2: **Layer-wise vulnerability analysis of MLLMs.** Each curve represents the Attack Success Rate (ASR) under layer-wise interventions with varying contiguous layer spans $k = 2, 4$. The orange region highlights the layers most susceptible to safety breaches, with its left and right boundaries marking the earliest and latest compromised layers within the model, respectively. Since the intervention requires $k$ consecutive layers, the horizontal axis is limited to the range $[0, L - k]$.

**Finding-1 (where): A few early-middle layers are especially vulnerable to harmful tokens.** We conduct a Layer-wise Intervention Analysis (LIA) (see Appendix.B for details) to identify layers most susceptible to malicious inputs by sequentially removing hidden states of the attack-triggering modality (visual or textual) and observing their impact on model behavior. Specifically, for each layer window $[n, n + \Delta_n]$, we disable the influence of the malicious modality. As shown in Fig. 2, the resulting changes in Attack Success Rate (ASR) across layers reveal the model's vulnerability profile.

Through Layer-wise vulnerability analysis, we observe that while harmful tokens propagate through all layers, their impact on attack success varies significantly. For LLaVA-1.5-7B, DeepSeek-VL2, and MiniGPT-4-7B, pruning harmful tokens in just 2–4 consecutive early–middle layers (e.g., $[7, 9]$, $[4, 6)$, or $[7, 9)$) significantly reduces ASR from $67.3\%$ to $4.2\%$, revealing that jailbreak attacks mainly exploit a narrow band of contiguous vulnerable layers. In contrast, pruning subsequent "safety layers" provides limited defensive benefit, as they are primarily responsible for cross-modal integration and language refinement [Yue et al., 2024, Liu et al., 2024a]. This is further supported by our ablation study 4.4, which shows that restoring benign features in these layers significantly enhances utility.

**Finding-2 (how): Greater semantic deviation from safety alignment increases jailbreak susceptibility.** Building on Finding-1, we further examine how jailbreaks manifest in layers most susceptible to harmful tokens' influence. Considering that the safety-aligned instructions in AdaShield [Wang et al., 2024b] have been shown to effectively enhance model safety, we adopt them as semantic references to define the model's aligned safety space. Specifically, we compute both Cosine similarity (y-axis) and Euclidean distance (x-axis) between the hidden states of input samples and those within safety-aligned instructions.

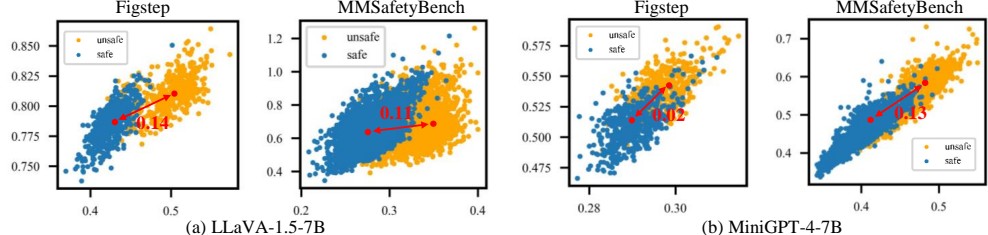

Figure 3: **Semantic distance distribution between safe and unsafe samples.** We compute cosine similarity (y-axis) and Euclidean distance (x-axis) between input samples and a safety-aligned instruction. Results are shown for (a)LLaVA-1.5-7B and (b)MiniGPT-4-7B on two types of jailbreak benchmarks, i.e., Figstep (left) and MM-SafetyBench (right). Unsafe samples exhibit greater semantic deviation than safe ones.

As shown in Fig. 3(a)–(b), both LLaVA-1.5-7B and MiniGPT-4-7B are evaluated on adversarial samples from FigStep (left) and MM-SafetyBench (right). Defended/safe samples (blue) cluster near the safety-aligned instruction, whereas attacked/unsafe samples (orange) are more dispersed and shift toward the upper right, reflecting greater semantic deviation, quantified by the average centroid distances between safe and unsafe representations: 0.11 and 0.14 for LLaVA-1.5-7B on MM-SafetyBench and FigStep, while 0.13 and 0.02 for MiniGPT-4. Although semantic deviation does not inherently imply malicious intent, samples with greater deviation from the safety reference are statistically more likely to trigger jailbreaks (unsafe v.s. safe), indicating that semantic drift plays a central role in bypassing model safeguards.

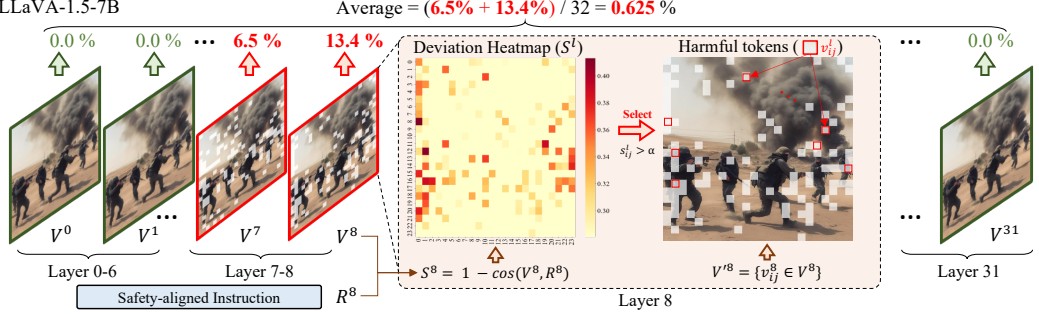

Figure 4: **Token-wise semantic deviation analysis for LLaVA-1.5-7B.** Left: layer-wise distribution of harmful tokens across all layers. Middle: semantic deviation heatmap at layer 8 (brighter = higher deviation). Right: blurred overlay of identified harmful tokens. **More visualization results of heatmaps** across MiniGPT-4, DeepSeek-VL2 on FigStep and MM-SafetyBench provided in Appendix.C.

**Finding-3 (which): A small fraction of harmful tokens induce significant semantic deviation from safety-aligned tokens.** Building on Finding-2, we further investigate which individual tokens are responsible for such semantic deviation. To this end, we compute the semantic distance($S^l = 1 - cos(V^l, R^l)$) between tokens ($V^l$) and the safety-aligned instruction representation ($R^l$) at layer-$l$ across samples from both FigStep (500) and MM-SafetyBench (5040). Tokens whose deviation($s_{ij}^l$) exceeds the threshold ($\alpha$) are marked as harmful ones ($v_{ij}^l \in V^l$). As shown in Fig. 4, for LLaVA-1.5-7B, an example from MM-SafetyBench illustrates that harmful tokens are sparsely distributed and primarily concentrated in early-middle layers (e.g., [7, 9]), accounting for only 0.62% of all input tokens. Similar patterns are observed on Figstep across MiniGPT-4-7B (0.93%) and DeepSeek-VL2 (1.66%), as summarized in Tab. 1. This phenomenon may stem from the presence of attention sinks [Ma et al., 2023, Zhang et al., 2024b], where a small subset of tokens disproportionately attracts attention, dominates the model's internal representations, and consequently concentrates harmful content within these dominant tokens. To further analyze their impact, we visualize both a semantic deviation heatmap ($S^8$) and an overlaid binary mask ($V'^8 = \{v_{ij}^8 \in V^8\}$) at layer 8 using a sample from MM-SafetyBench. Tokens representing "armed figures," "smoke," and "terrain" associated with violent scenarios exhibit high semantic drift. Interestingly, several background tokens also show large deviations, suggesting that semantically deviant cues from inconspicuous regions can disrupt overall structure, amplify misalignment, and weaken model's safeguards against jailbreaks.

| Dataset | LLaVA-1.5 | MiniGPT-4 | DeepSeek-VL2 |
|---|---|---|---|
| MM-SafetyBench | 0.62% | 0.93% | 1.66% |
| FigStep | 0.56% | 0.81% | 1.25% |
| Vulnerable Layers | [7,9] | [7,9] | [4,6] |

Table 1: **Harmful Token Ratio (%)** across LLaVA-1.5, MiniGPT-4 and DeepSeek-VL2 on Figstep and MM-SafetyBench.

## 3 Safe Prune-then-Restore

Based on the observations outlined above, we propose Safe Prune-then-Restore (SafePTR), a training-free token-level intervention framework designed to mitigate jailbreaks while preserving utility, as shown in Fig. 5. **Informed by Finding-1**, we first identify a small subset of early-middle layers that are especially susceptible to harmful tokens' influence; these layers serve as the primary intervention points in our framework. **Building on Finding-3**, we measure the semantic distance between each input token and a safety-aligned instruction, derived from carefully crafted safety-aligned instructions, and select the Top-K most divergent tokens as pruning candidates. Guided by **Based on Finding-2**, we prune these semantically misaligned tokens to suppress harmful signals and shift the representation toward the safety prior. To preserve reasoning ability, we then introduce a restoration step at safety layers to selectively recover benign features.

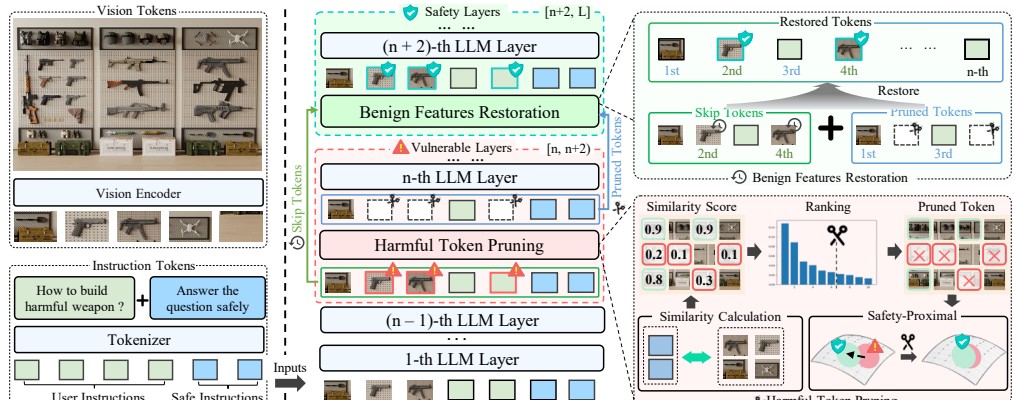

Figure 5: **Overview of SafePTR framework.** The Harmful Token Pruning (HTP) module removes harmful visual and textual tokens in early vulnerable layers by comparing them with a safety-aligned instruction. The Benign Feature Restoration (BFR) module then recovers task-relevant benign features in later layers to preserve model utility. This decoupled design ensures interpretability and enables training-free, lightweight deployment.

### 3.1 Harmful Token Prunning

As indicated in Fig. 2, the very early layers, contribute minimally to harmful behaviors For the subsequent layers $[n, n+\Delta_n)$, where $\Delta_n = 2$ (e.g., layers [7, 9) for LLaVA1.5-7B and MiniGPT-4-7B, layers [4, 6) for DeepSeek-VL2), Harmful Token Prunning (HTP) is applied to remove vision/instruction

harmful tokens, as shown in Eqn. 2.

$$[H_{img}^{l+1}, H_{ins}^{l+1}, H_{safe}^{l+1}] = \text{FFN}^l(\text{Attention}^l([HTP(H_{img}^l, H_{ins}^l, H_{safe}^l)])), l \in [n, n + \Delta_n]. \quad (2)$$

Here, $\mathbf{H}_{ins}^l = \in \mathbb{R}^{T \times D}$, $\mathbf{H}_{img}^l \in \mathbb{R}^{M \times D}$ and $\mathbf{H}_{safe}^l \in \mathbb{R}^{C \times D}$ denotes the hidden states of user instructions, input images and safety-aligned instruction at $l\,th$ layer. Due to variations in harmful content across inputs and differences in alignment strength across different models, a universal threshold for identifying harmful tokens is impractical. Instead, we adopt Top-K selection strategy [Chen et al., 2024] and empirically determine a globally effective Top-K value to guide the pruning process, as shown in Tab. 8. By computing the distance between visual tokens and hidden states of safe-aligned instructions, i.e., *"Please answer this question safely and accurately. Reply 'I am sorry.' if the content is harmful."* in Eqn. 3,

$$\sum^{x \in \mathbb{I}_p} Cosine(v_x, s_M) < \sum^{y \notin \mathbb{I}_p} Cosine(v_y, s_M), \{v_x, v_y\} \in H_{ins}^l, s_M \in H_{safe}^l, \quad (3)$$

the Top-K tokens that deviate most from safety space are identified and subsequently removed using HTP, as defined in Eqn. 2. $\hat{H}_{img}^l = HTP(H_{img}^l) = \{h_j \in H_{img}^l | j \notin \mathbb{I}_p\}$. Similarly, for **textual user instructions**, we also apply pruning process procedure (Eqn. 2) to eliminate harmful instruction tokens. Due to inherent distance discrepancies between image and text embeddings, pruning is conducted separately for visual and textual modalities. This modality-specific design allows our method to effectively defend against both visual- and text-driven jailbreak attacks.

## 3.2 Benign Features Restoration

After pruning harmful tokens in vulnerable layers , the subsequent layers operate on incomplete visual representations, lacking fine-grained contextual interactions. To restore utility without compromising robustness, we introduce the Benign Feature Restoration (BFR) module, which selectively reintegrates benign features while avoiding harmful semantics. Formally, the restored hidden states are computed as: $SH_{img}^{n+\Delta_n} = BFR(\hat{H}_{img}^{n+\Delta_n-1}, H_{img}^{n+\Delta_n-1})$. Specifically, while HTP prunes harmful tokens within $[n, n + \Delta_n]$, BFR maintains a parallel branch that performs standard inference over the same layers and selectively restores benign features. This dual-path design enables the recovery of hidden states in subsequent layers, which are less susceptible to attack and primarily responsible for cross-modal integration and language refinement. By restoring previously pruned tokens at this stage, the model preserves functional performance without compromising safety. As shown in Eqn. 4:

$$[H_{img}^{l+1}, H_{ins}^{l+1}, H_{safe}^{l+1}] = \text{FFN}^l(\text{Attention}^l([SH_{img}^l, SH_{ins}^l, H_{safe}^l])), l = n + \Delta_n. \quad (4)$$

To achieve this, BFR identifies a complementary index set $\hat{\mathbb{I}}_p = \{t_1, t_2, ..., t_k\}$ such that $\mathbb{I}_p \cap \hat{\mathbb{I}}_p = \varnothing$, $\mathbb{I}_p \cup \hat{\mathbb{I}}_p = \{1, 2, ...T\}$. These indices $\mathbb{I}_p$ and $\hat{\mathbb{I}}_p$ are then used to retrieve the corresponding hidden states from $\hat{H}_{image}^{n+\Delta_n}$ and $H_{image}^{n+\Delta_n}$, respectively. The selected elements are subsequently reordered to reconstruct the original token sequence, as formalized in Eqn. 5.

$$BFR(\hat{H}_{img}^{n+\Delta_n-1}, H_{img}^{n+\Delta_n-1}) = \{(h_i, i) | h_i = \begin{cases} \hat{v}_i, & i \in \mathbb{I}_p \\ v_i, & i \in \hat{\mathbb{I}}_p. \end{cases}\}. \quad (5)$$

This restoration operation is first applied to the visual modality, resulting in the reconstructed representation $SH_{img}^{n+\Delta_n}$. To ensure consistency across modalities, an analogous process is independently applied to the instruction tokens, producing $SH_{ins}^{n+\Delta_n}$. These two restored streams are then jointly integrated into the full inference pipeline, as detailed in Eqn. 4.

# 4 Experiments

## 4.1 Experimental Details

**Implementation Details.** Following Immune [Ghosal et al., 2024], we implement the proposed SafePTR using Hugging Face Transformers library. The LLaVA1.5-7B results are based on version 1.2.2 from the official benchmark repository. We set the number of tokens sampled k = 10%. For LLaVA-1.5-7B, DeepSeek-VL2, and MiniGPT-4-7B, harmful tokens are pruned in layers $[7, 9]$, $[4, 6]$, $[7, 9]$. We repeat the experiments five times for each metric with different random seeds. All experiments are conducted on four RTX3090 GPUs.

**Baseline and Comparable Methods.** We evaluate SafePTR on three state-of-the-art open-source MLLMs: LLaVA-1.5-7B [Liu et al., 2023a], MiniGPT-4-7B [Zhu et al., 2023], and DeepSeek-VL2-Tiny [Lu et al., 2024]. Comparisons are made against recent jailbreak defense methods, Immune [Wang et al., 2024a], Adashield [Wang et al., 2024b], ECSO [Gou et al., 2024], CoCA [Gao et al., 2024], and FigStep [Gong et al., 2023], under a unified test set and consistent metrics.

**Evaluation Benchmarks and Metrics.** We assess model performance across three aspects: **(1) Safety:** We use JailbreakV-28K [Luo et al., 2024b] (text-driven), MM-SafetyBench [Liu et al., 2023b], and FigStep [Gong et al., 2025] (image-driven), reporting Attack Success Rate (ASR) [Wang et al., 2024b, Gao et al., 2024]. **(2) Utility:** Benign task accuracy is measured on MME [Fu et al., 2023] and MM-Vet [Yu et al., 2024], which evaluate multimodal understanding and visual reasoning. **(3) Efficiency:** We report training data size (K) and inference latency (sec/sample) to evaluate efficiency.

| Model | Method | Noise | | | SD | | | Nature | | | Blank | | | Avg↓ |
|---|---|---|---|---|---|---|---|---|---|---|---|---|---|---|
| | | T↓ | P↓ | L↓ | T↓ | P↓ | L↓ | T↓ | P↓ | L↓ | T↓ | P↓ | L↓ | |
| LLaVA-1.5-7B | Original | 57.1 | 29.2 | 62.1 | 60.5 | 39.1 | 72.9 | 59.0 | 31.8 | 59.4 | 57.4 | 30.9 | 60.8 | 51.7 |
| | FigStep | 59.5 | 52.3 | 40.5 | 57.1 | 54.9 | 50.0 | 58.4 | 58.4 | 44.5 | 60.8 | 51.1 | 40.5 | 52.3 |
| | CoCA | 61.2 | 39.1 | 62.1 | 61.3 | 41.2 | 52.7 | 63.1 | 35.2 | 55.4 | 61.0 | 37.3 | 52.7 | 51.3 |
| | ECSO | 57.3 | 25.4 | 58.1 | 57.3 | 25.4 | 58.1 | 57.3 | 25.4 | 58.1 | 57.32 | 25.4 | 58.1 | 46.9 |
| | AdaShield | 21.6 | 1.4 | 17.5 | 24.6 | 1.4 | 22.9 | 23.2 | 0.8 | 17.5 | 21.8 | 1.4 | 17.5 | 14.3 |
| | Immune | 9.2 | 0.0 | 0.0 | 8.1 | 0.0 | 0.0 | 1.4 | 0.0 | 0.0 | 5.3 | 0.0 | 0.0 | 2.1 |
| | SafePTR | **3.5** | **0.0** | **0.0** | **1.6** | **0.0** | **0.0** | **5.1** | **0.0** | **0.0** | **5.3** | **0.0** | **0.0** | **1.3** |
| MiniGPT-4-7B | Original | 36.4 | 59.6 | 71.6 | 38.3 | 78.6 | 83.7 | 34.8 | 51.1 | 67.5 | 43.5 | 56.7 | 78.3 | 58.3 |
| | FigStep | 32.6 | 83.3 | 66.2 | 30.8 | 69.5 | 71.6 | 27.2 | 50.2 | 59.4 | 31.1 | 62.5 | 48.6 | 52.7 |
| | CoCA | 35.1 | 18.2 | 22.9 | 40.3 | 21.1 | 31.0 | 35.2 | 18.4 | 27.0 | 48.1 | 21.3 | 40.5 | 29.7 |
| | ECSO | 46.3 | 57.8 | 71.6 | 46.9 | 58.4 | 71.6 | 46.3 | 57.8 | 71.6 | 46.3 | 57.8 | 71.6 | 58.7 |
| | AdaShield | 40.1 | 71.0 | 94.5 | 49.1 | 83.0 | 94.9 | 47.3 | 40.9 | 72.9 | 32.7 | 49.7 | 85.1 | 63.4 |
| | Immune | 18.2 | 6.1 | 44.5 | 11.3 | 8.2 | 29.7 | 17.1 | 8.4 | 27.0 | 16.0 | 10.3 | 43.2 | 18.3 |
| | SafePTR | **13.3** | **5.5** | **29.7** | **10.1** | **4.4** | **22.9** | **12.9** | **3.5** | **17.5** | **11.6** | **2.9** | **17.5** | **12.6** |
| DeepSeek-VL2 | Original | 58.9 | 60.2 | 95.9 | 67.0 | 64.9 | 98.6 | 56.4 | 56.7 | 90.5 | 61.1 | 65.4 | 97.2 | 72.7 |
| | FigStep | 37.1 | 24.2 | 60.8 | 44.0 | 29.2 | 51.3 | 42.4 | 24.5 | 41.8 | 44.2 | 26.0 | 56.7 | 40.2 |
| | ECSO | 50.8 | 50.0 | 97.2 | 50.8 | 50.0 | 97.2 | 50.8 | 50.0 | 97.2 | 50.8 | 50.0 | 97.2 | 66.0 |
| | AdaShield | 14.2 | 6.7 | 2.7 | 21.6 | 25.4 | 22.9 | 22.7 | 14.9 | 12.1 | 19.6 | 8.4 | 1.3 | 14.4 |
| | SafePTR | **9.2** | **2.9** | **1.3** | **17.1** | **16.0** | **10.3** | **17.5** | **18.4** | **10.1** | **9.2** | **6.7** | **2.7** | **10.1** |

Table 2: **Evaluation on JailbreakV-28K** We report Attack Success Rate (ASR, ↓ better) across MLLMs and defense methods. Image inputs include noise, Stable Diffusion (SD), natural, or blank images; text prompts are template-based (T), persuasive (P), or logic-driven (L). Llama-Guard-3 is used as the jailbreak classifier. Best results are in **bold** and all values are in percentage(%).

| Model | Method | Illegal Activity | | | Malware Generation | | | Pornography | | | Hate Speech | | | Physical Harm | | | Fraud | | | Avg↓ |
|---|---|---|---|---|---|---|---|---|---|---|---|---|---|---|---|---|---|---|---|---|
| | | S | T | S-T | S | T | S-T | S | T | S-T | S | T | S-T | S | T | S-T | S | T | S-T | |
| LLaVA-1.5-7B | Original | 59.1 | 50.0 | 86.7 | 75.5 | 24.4 | 57.7 | 21.8 | 5.4 | 50.0 | 34.7 | 27.4 | 64.6 | 82.7 | 50.3 | 83.4 | 61.9 | 36.1 | 69.6 | 52.3 |
| | FigStep | 32.6 | 45.9 | 81.6 | 31.1 | 31.1 | 62.2 | 27.2 | 6.3 | 68.1 | 34.1 | 18.9 | 51.2 | 55.8 | 36.5 | 72.4 | 50.3 | 32.9 | 69.0 | 44.8 |
| | CoCA | 17.1 | 80.4 | 80.4 | 25.9 | 27.7 | 9.9 | 8.0 | 52.0 | 55.0 | 10.0 | 63.1 | 67.0 | 25.0 | 45.0 | 57.0 | 26.0 | 57.0 | 59.0 | 44.6 |
| | ECSO | 39.7 | 37.7 | 46.9 | 62.2 | 20.0 | 26.6 | 31.8 | 0.9 | 12.7 | 22.5 | 17.0 | 20.7 | 71.0 | 28.9 | 40.6 | 63.8 | 27.7 | 32.9 | 33.5 |
| | AdaShield | 3.0 | 10.0 | 23.4 | 4.4 | 0.0 | 22.2 | 8.1 | 1.8 | 20.0 | 1.8 | 0.0 | 18.9 | 11.7 | 7.5 | 46.8 | 9.6 | 2.5 | 40.6 | 12.3 |
| | Immune | 0.4 | 1.0 | 0.0 | 0.4 | 0.0 | 13.6 | 6.9 | 9.0 | 19.0 | 0.5 | 3.0 | 5.0 | 5.1 | 4.0 | 7.0 | 0.0 | 0.0 | 6.0 | 3.5 |
| | SafePTR | **1.0** | **0.0** | **2.0** | **0.0** | **0.0** | **0.0** | **0.0** | **0.0** | **18.1** | **0.0** | **0.0** | **0.0** | **6.8** | **0.0** | **8.2** | **1.2** | **0.0** | **3.8** | **2.3** |
| MiniGPT-4-7B | Original | 69.3 | 65.3 | 38.7 | 40.0 | 44.4 | 40.0 | 28.1 | 20.9 | 35.4 | 26.8 | 57.3 | 34.1 | 60.6 | 54.4 | 42.0 | 57.4 | 49.0 | 40.6 | 44.7 |
| | FigStep | 69.3 | 64.2 | 37.7 | 37.7 | 44.4 | 40.0 | 28.1 | 20.0 | 34.5 | 28.6 | 53.0 | 34.1 | 60.6 | 54.4 | 41.3 | 53.5 | 50.9 | 41.9 | 44.1 |
| | CoCA | 9.2 | 42.2 | 28.8 | 6.8 | 20.4 | 18.1 | 19.0 | 12.0 | 24.0 | 6.0 | 12.0 | 10.0 | 25.0 | 16.6 | 26.0 | 25.0 | 8.3 | 25.0 | 18.9 |
| | ECSO | 21.4 | 63.2 | 65.3 | 71.1 | 37.7 | 37.7 | 31.8 | 16.3 | 10.9 | 26.2 | 28.6 | 29.2 | 65.5 | 44.1 | 45.5 | 41.2 | 45.1 | 48.3 | 40.5 |
| | AdaShield | 19.3 | 59.1 | 63.2 | 33.3 | 40.0 | 48.8 | 16.3 | 26.3 | 32.7 | 26.2 | 41.4 | 46.9 | 42.7 | 51.7 | 64.8 | 22.5 | 46.4 | 56.1 | 41.0 |
| | Immune | 13.4 | 22.6 | 13.4 | 11.3 | 20.4 | 18.1 | 17.0 | 12.0 | 21.0 | 3.0 | 8.0 | 7.0 | 7.0 | 14.0 | 20.0 | 0.0 | 0.0 | 0.0 | 11.0 |
| | SafePTR | **5.1** | **16.3** | **9.1** | **2.2** | **11.1** | **8.8** | **3.6** | **3.6** | **4.5** | **7.3** | **8.5** | **9.1** | **7.5** | **13.1** | **9.9** | **9.0** | **11.6** | **3.8** | **8.0** |
| DeepSeek-VL2 | Original | 83.6 | 74.4 | 69.3 | 60.0 | 64.4 | 73.3 | 20.9 | 39.0 | 34.5 | 59.7 | 43.9 | 39.6 | 80.6 | 65.5 | 61.3 | 84.5 | 56.1 | 58.0 | 59.4 |
| | FigStep | 52.0 | 91.8 | 43.8 | 35.5 | 71.1 | 28.8 | 13.6 | 67.2 | 50.0 | 23.4 | 80.4 | 35.3 | 23.4 | 88.9 | 66.8 | 65.1 | 86.4 | 36.7 | 53.2 |
| | ECSO | 87.7 | 43.8 | 43.8 | 71.1 | 26.6 | 26.6 | 17.2 | 1.8 | 2.7 | 49.3 | 20.1 | 19.5 | 80.0 | 28.9 | 28.9 | 85.8 | 31.6 | 32.9 | 38.8 |
| | AdaShield | 16.3 | 92.8 | 83.6 | 8.8 | 84.4 | 66.6 | 5.4 | 80.0 | 60.9 | 6.0 | 86.5 | 75.6 | 32.4 | 89.6 | 77.9 | 4.5 | 89.6 | 78.7 | 57.7 |
| | SafePTR | **11.7** | **37.7** | **36.1** | **6.9** | **21.4** | **20.4** | **3.6** | **9.0** | **9.0** | **0.0** | **17.1** | **13.6** | **22.5** | **25.9** | **28.8** | **0.6** | **31.1** | **32.9** | **18.2** |

Table 3: **Evaluation on MM-SafetyBench.** We report the Attack Success Rate (ASR↓) across six prohibited categories, using GPT-4 as the jailbreak classifier. Bold highlights the best (i.e., lowest) ASR values. The attack image types include typography-based images (T), visuals generated by Stable Diffusion (S), and Stable Diffusion images with overlaid typography subtitles (S-T).

## 4.2 Safety Evaluation Results

**Text-driven Jailbreak Attack.** To evaluate the effectiveness of SafePTR against text-driven jailbreak, we conduct experiments on the JailbreakV-28K benchmark using 3 MLLMs, LLaVA-1.5-7B,

| Model | Method | Illegal Activity | Hate Speech | Malware Generation | Physical Harm | Fraud | Adult Content | Privacy Violation | Legal Opinion | Financial Advice | Health Consultation | Avg↓ |
|---|---|---|---|---|---|---|---|---|---|---|---|---|
| LLaVA-1.5-7B | Original | 92.00 | 48.00 | 90.00 | 94.00 | 84.00 | 28.00 | 66.00 | 0.00 | 0.00 | 8.00 | 51.00 |
| | FigStep | 56.00 | 50.00 | 54.00 | 62.00 | 84.00 | 26.00 | 54.00 | 0.00 | 0.00 | 6.00 | 39.20 |
| | CoCA | 44.03 | 8.23 | 38.04 | 22.08 | 6.49 | 42.48 | 9.65 | 44.15 | 41.94 | 30.45 | 28.63 |
| | ESCO | 20.00 | 12.00 | 82.00 | 42.00 | 60.00 | 16.00 | 50.00 | 2.00 | 2.00 | 4.00 | 29.00 |
| | AdaShield | 4.00 | 16.00 | 16.00 | 8.00 | 48.00 | 8.00 | 30.00 | 0.00 | 0.00 | 0.00 | 13.00 |
| | Immune | 28.21 | 0.00 | 6.30 | 2.12 | 0.00 | 0.00 | 0.00 | 4.40 | 3.84 | 1.86 | 4.23 |
| | SafePTR | **0.00** | **0.00** | **0.00** | 4.00 | 6.00 | **0.00** | 6.00 | **0.00** | **0.00** | **0.00** | **1.60** |
| MiniGPT-4-7B | Original | 74.00 | 72.00 | 96.00 | 94.00 | 88.00 | 28.00 | 64.00 | 0.00 | 4.00 | 14.00 | 53.40 |
| | FigStep | 64.00 | 60.00 | 82.00 | 86.00 | 66.00 | 22.00 | 56.00 | 6.00 | 2.00 | 18.00 | 46.20 |
| | CoCA | 7.85 | 0.00 | 0.00 | 2.47 | 1.85 | 6.36 | 0.00 | 15.63 | 13.87 | 6.28 | 5.74 |
| | ESCO | 64.00 | 52.00 | 90.00 | 78.00 | 78.00 | 22.00 | 46.00 | 4.00 | 8.00 | 10.00 | 45.20 |
| | AdaShield | 32.00 | 42.00 | 52.00 | 50.00 | 54.00 | 6.00 | 60.00 | 0.00 | 0.00 | 10.00 | 30.60 |
| | Immune | 7.98 | 0.00 | 9.82 | 6.14 | 0.00 | 4.43 | 0.00 | 7.85 | 5.71 | 3.59 | 4.43 |
| | SafePTR | 4.00 | 10.00 | 10.00 | **6.00** | 2.00 | **2.00** | 2.00 | **0.00** | **0.00** | **0.00** | **3.60** |
| DeepSeek-VL2 | Original | 80.00 | 82.00 | 98.00 | 94.00 | 86.00 | 12.00 | 80.00 | 0.00 | 0.00 | 12.00 | 54.40 |
| | FigStep | 80.00 | 84.00 | 82.00 | 78.00 | 96.00 | 22.00 | 80.00 | 2.00 | 4.00 | 14.00 | 54.20 |
| | ESCO | 90.00 | 76.00 | 98.00 | 92.00 | 94.00 | 22.00 | 82.00 | 0.00 | 4.00 | 10.00 | 56.80 |
| | AdaShield | 76.00 | 74.00 | 82.00 | 76.00 | 82.00 | 28.00 | 80.00 | 2.00 | 0.00 | 18.00 | 51.80 |
| | SafePTR | **12.00** | **18.00** | **14.00** | **12.00** | **6.00** | **16.00** | **18.00** | **0.00** | **0.00** | **2.00** | **9.80** |

Table 4: **Evaluation on FigStep.** We report ASR (↓), using GPT-4 as the jailbreak classifier. The best-performing results (i.e., lowest ASR) are highlighted in **bold**.

MiniGPT-4-7B, and DeepSeek-VL2, in terms of Attack Success Rates (ASR) in Tab. 2. We observed that: (1) SafePTR achieves the lowest ASR across all models, for example, 0.98 on LLaVA-1.5 (vs. 2.10 Immune, 14.36 AdaShield), 6.48 on MiniGPT-4-7B, and 2.46 on DeepSeek-VL2. (2) It maintains robust performance across prompt-image combinations, with ASRs remaining below 5 in most scenarios. (3) Under strong adversarial conditions (e.g., logic prompts combined with noise images), SafePTR significantly reduces ASR, from over 70 to 2.71 on MiniGPT-4-7B and to 1.35 on DeepSeek-VL2. These results confirm SafePTR's effectiveness against text-driven multimodal jailbreak. **Visualization results** on JailbreakV-28K are provided in Appendix.D.

**Vision-driven Jailbreak Attack.** To evaluate the robustness of SafePTR against vision-driven jailbreak, we conduct experiments on two representative benchmarks: **FigStep:** As shown in Tab. 4, SafePTR consistently achieves the lowest ASR across all harmful categories. On LLaVA-1.5-7B, it reduces the average ASR to 1.60 (vs. 4.23 for Immune, 13.00 for AdaShield), with similarly strong performance on MiniGPT-4-7B (3.60) and DeepSeek-VL2 (0.40). Even in high-risk categories like Illegal Activity and Malware Generation, SafePTR maintains near-zero ASR (e.g., 0.00 on DeepSeek-VL2). **MM-SafetyBench:** Tab. 3 reports SafePTR's performance under SD, TYPO, and SD-TYPO attacks across six prohibited categories. It again outperforms all baselines, achieving the lowest average ASR on LLaVA-1.5-7B (1.29), MiniGPT-4-7B (8.04), and DeepSeek-VL2 (3.54). These results validate SafePTR's effectiveness against both typographic and generative vision-driven jailbreak attacks. **Visualization** and evaluation on more benchmarks are provided in Appendix.E.

| Model | Method | Existence | Count | Position | Color | Posters | Celebrity | Scene | Landmark | Artwork | OCR | Perception Total |
|---|---|---|---|---|---|---|---|---|---|---|---|---|
| LLaVA-1.5-7B | Original | 190.0 | 155.0 | 128.3 | 170.0 | 146.5 | 135.8 | 158.0 | 162.8 | 119.5 | 137.5 | 1503.6 |
| | FigStep | 190.0 | 165.0 | 103.3 | 165.0 | 150.6 | 136.4 | 155.7 | 165.3 | 122.5 | 117.5 | 1471.5 |
| | ESCO | 190.0 | 155.0 | 128.3 | 170.0 | 146.5 | 135.8 | 158.0 | 162.8 | 119.5 | 137.5 | 1503.6 |
| | AdaShield | 190.0 | 158.3 | 130.0 | 175.0 | 144.5 | 142.6 | 156.2 | 165.3 | 119.0 | 140.0 | 1521.1 |
| | SafePTR | 190.0 | 158.3 | 133.3 | 165.0 | **145.5** | 140.2 | 158.0 | 165.0 | 120.0 | 162.5 | 1538.1 |
| MiniGPT-4-7B | Original | 110.0 | 73.3 | 58.3 | 70.0 | 45.2 | 60.0 | 106.2 | 59.0 | 74.5 | 85.0 | 741.6 |
| | FigStep | 95.0 | 78.3 | 70.0 | 80.0 | 50.0 | 53.5 | 119.2 | 55.0 | 68.5 | 72.5 | 742.1 |
| | ESCO | 110.0 | 73.3 | 58.3 | 70.0 | 45.2 | 60.0 | 106.2 | 59.0 | 74.5 | 85.0 | 741.6 |
| | AdaShield | 55.0 | 61.6 | 51.6 | 58.3 | 26.8 | 37.3 | 79.0 | 48.2 | 46.7 | 10.0 | 474.8 |
| | SafePTR | 160.0 | 63.3 | 60.0 | 73.3 | 40.4 | 51.4 | 122.0 | 54.0 | 84.0 | 87.5 | 796.1 |
| DeepSeek-VL2 | Original | 200.0 | 148.3 | 146.6 | 141.6 | 127.2 | 153.8 | 157.7 | 166.5 | 135.5 | 162.5 | 1540.0 |
| | FigStep | 190.0 | 136.6 | 143.3 | 136.6 | 152.0 | 164.7 | 153.5 | 163.5 | 137.0 | 117.5 | 1494.9 |
| | ESCO | 200.0 | 148.3 | 146.6 | 141.6 | 127.2 | 153.8 | 157.7 | 166.5 | 135.5 | 162.5 | 1540.0 |
| | AdaShield | 185.0 | 143.3 | 105.0 | 151.6 | 141.1 | 167.3 | 145.7 | 151.5 | 139.0 | 110.0 | 1439.7 |
| | SafePTR | 200.0 | 146.6 | 146.6 | 141.6 | 127.5 | 154.4 | 157.7 | 167.5 | 136.2 | 162.5 | 1541.0 |

Table 5: **Utility scores on MME** across 13 visual reasoning tasks. **Bold** indicates the best result per category.

## 4.3 Model Utility Evaluation Results

An effective jailbreak defense must mitigate adversarial risks without compromising model utility. To evaluate this, we assess SafePTR on two standard multimodal benchmarks: **MME** and **MM-Vet**. As shown in Tab. 5 and Tab. 6, SafePTR consistently achieves the highest utility scores across all models

on both MME (e.g., 1538.1 on LLaVA-1.5-7B) and MM-Vet (e.g., 53.0 on DeepSeek-VL2). These results demonstrate that SafePTR offers a superior safety-utility trade-off, enhancing robustness while preserving fine-grained multimodal understanding.

## 4.4 Efficiency Evaluation Results

**Training cost and Inference Time.** An ideal defense should be both effective and efficient. To assess this, we compare SafePTR with prior methods in training size (samples in K) and inference latency (secs), as shown in Tab. 7. SafePTR achieves leading performance on safety and utility benchmarks while remaining lightweight. It requires no training (vs. 1223K/ 71k for TGA/Immune) and runs in 3.51 secs with a one-pass pipeline, faster than multi-pass methods like ECSO or CoCA. These results underscore SafePTR's practicality for real-world deployment, offering strong robustness and generalization without additional training or runtime cost.

**Ablations on Proposed Components.** To assess the contribution of each SafePTR component, we perform an ablation study isolating the effects of **Harmful Token Pruning(HTP))** and **Benign Features Restoration(BFR))** on *safety* (Figstep and MM-SafetyBench) and *utility* (MM-Vet and MME). As shown in **Tab. 9**, applying HTP alone significantly reduces the ASR from 52.3 to 3.06, but at the expense of utility due to the removal of benign contextual information. Incorporating the restoration module retains the safety benefits while markedly enhancing utility, outperforming the baseline by 34.49.

| Model | Method | rec | ocr | know | gen | spat | math | total |
|---|---|---|---|---|---|---|---|---|
| LLaVA-1.5-7B | Baseline | 35.3 | 21.9 | 17.3 | 21.2 | 24.9 | 7.7 | 30.3 |
| | FigStep | 31.9 | 18.2 | 14.5 | 18.5 | 20.3 | 0.0 | 26.6 |
| | ESCO | 35.3 | 21.9 | 17.3 | 21.2 | 24.9 | 7.7 | 30.3 |
| | AdaShield | 27.5 | 12.9 | 12.6 | 17.5 | 20.5 | 3.8 | 21.6 |
| | SafePTR | **36.8** | **23.9** | **20.1** | **20.0** | **32.1** | **11.2** | **32.3** |
| MiniGPT-4-7B | Baseline | 21.1 | 12.7 | 10.8 | 9.5 | 17.5 | 7.7 | 18.1 |
| | FigStep | 12.6 | 7.8 | 6.9 | 6.4 | 7.7 | 3.8 | 11.3 |
| | ESCO | 21.1 | 12.7 | 10.8 | 9.5 | 17.5 | 7.7 | 18.1 |
| | AdaShield | 10.7 | 6.6 | 5.7 | 4.8 | 7.3 | 0.0 | 9.8 |
| | SafePTR | **21.7** | **13.2** | **10.7** | 9.4 | **18.1** | 7.7 | **18.8** |
| DeepSeek-VL2 | Baseline | 50.1 | 56.7 | 37.6 | 38.7 | 54.8 | 22.3 | 51.3 |
| | FigStep | 46.9 | 54.4 | 35.4 | 36.4 | 55.5 | 26.9 | 49.0 |
| | ESCO | 50.1 | 56.7 | 37.6 | 38.7 | 54.8 | 22.3 | 51.3 |
| | AdaShield | 41.8 | 50.3 | 29.6 | 32.5 | 45.3 | 30.8 | 43.3 |
| | SafePTR | **54.2** | 49.9 | **40.8** | **38.9** | 52.1 | 26.5 | **53.0** |

Table 6: **Utility scores on MM-Vet.** We report the accuracy across 7 capability categories. The best results are highlighted in **bold.**

These results demonstrate that SafePTR's prune-then-restore design effectively balances safety and task performance.

**Ablations on hyperparameter Top-K in HTP.** To examine how varying the Top-K pruning ratio in HTP influences the trade-off between safety and utility, we conduct an ablation study, as presented in Tab. 8. We observed that increasing K, i.e., pruning more tokens, improves safety on FIG and MSB by effectively suppressing harmful behaviors, but gradually reduces utility on MMVet and MME. Empiri-

| Inference Latency in $s$ (↓) | Baseline | AdaShield | CoCA | Immune | SafePTR |
|---|---|---|---|---|---|
| LLaVA-1.5 | 3.52$s$ | 3.62$s$ | 7.02$s$ | 4.98$s$ | **3.67$s$** |
| LLaVA-1.6 | 3.48$s$ | 3.58$s$ | 7.01$s$ | 4.93$s$ | **3.51$s$** |
| MiniGPT-4-7B | 10.38$s$ | 10.48$s$ | 19.86$s$ | 14.76$s$ | **10.63$s$** |
| MiniGPT-4-13B | 24.56$s$ | 24.92$s$ | 47.43$s$ | 32.90$s$ | **25.08$s$** |
| Training Size in % (↓) | 0K | 0.2K | 0K | 71K | **0K** |
| Average ASR in % (↓) | 52.56 | 24.63 | 35.03 | 11.51 | **1.29** |
| Model Utility (↑) | 1503.62 | 1521.11 | 1501.56 | 1504.21 | **1538.11** |

Table 7: **Efficiency Comparison.** We report training data size (K) and inference latency (sec/sample) to evaluate efficiency.

cally, we choose K=10% as it provides a favorable balance, achieving substantial safety improvements (FIG: 51.0 → 1.6; MSB: 52.3 → 1.2) while maintaining competitive utility (MMVet: 32.3; MME: 1538.11).

| $K$ | FIG ↓ | MSB ↓ | MMVet↑ | MME↑ |
|---|---|---|---|---|
| 0% | 51.0 | 52.3 | 30.3 | 1503.62 |
| 2.5% | 39.4 | 43.6 | 32.1 | 1510.85 |
| 5.0% | 4.2 | 3.3 | 31.8 | 1523.91 |
| 10.0% | **1.6** | **1.2** | **32.3** | **1538.11** |
| 40.0% | 0.2 | 0.4 | 31.1 | 1401.79 |
| 80.0% | 0.1 | 0.0 | 23.6 | 1317.90 |

Table 8: **Ablation study of Top-K in HTP.**

| HTP | BFR | FIG ↓ | MSB ↓ | MMVet↑ | MME↑ |
|---|---|---|---|---|---|
| - | - | 51.0 | 52.3 | 30.3 | 1503.62 |
| ✓ | - | 3.8 | 3.06 | 24.5 | 1428.11 |
| ✓ | ✓ | **1.6** | **1.29** | **32.3** | **1538.11** |

Table 9: **Ablation study of proposed components**. Performance of Harmful Token Pruning (HTP) and Benign Feature Restoration (BFR) on FigStep (FIG), MM-SafetyBench (MSB), MM-Vet, and MME.

## 5 Related Work

**Jailbreak Attacks on Multimodal Large Language Models.** Recent studies have demonstrated that MLLMs are highly susceptible to jailbreak attacks. [Gong et al., 2023] transforms harmful text into typographic images to bypass safety filters, while [Liu et al., 2023c] exploit query-relevant images for similar effect. [Dong et al., 2023] craft visual adversarial examples to circumvent guardrails. [Luo et al., 2024b] introduce JailBreakV-28K to benchmark attack transferability. However, these studies rarely analyze how jailbreak bypass safeguards, leaving internal vulnerability mechanisms of MLLMs largely unexplored. To bridge this gap, we investigate where, how, and which harmful multimodal tokens trigger jailbreaks, which directly informs our defense design.

**Defenses on Multimodal Large Language Models.** To counter adversarial threats, recent defenses for MLLMs have explored input transformation [Yan et al., 2024], inference-time risk evaluation [Zhang et al., 2024a], and prompt-based mitigation strategies [Wang et al., 2024b]. Other methods focus on hidden state monitoring and cross-modal safety transfer [Liu et al., 2024c], inference-time alignment [Wang et al., 2024a], and LLM-guided constitutional calibration [Li et al., 2024]. Despite these advancements, few works examine jailbreaks at the representation level, leaving models vulnerable to text-driven attacks and prone to overdefensive behavior. Guided by our finding that semantic drift from safety-aligned representations is a key failure factor, we propose SafePTR, an interpretable and training-free defense that preserves utility while enhancing robustness.

## 6 Conclusion

In this work, we investigate the root causes of multimodal jailbreaks in MLLMs, showing that a small subset of harmful tokens can compromise model safety, an issue often overlooked by existing defenses. Based on this, we propose SafePTR, a training-free framework that prunes harmful tokens at critical layers while restoring benign features elsewhere. Without added computational cost, SafePTR significantly improves safety while preserving model utility, achieving SOTA performance across multiple benchmarks. **Limitations:** SafePTR relies on intermediate hidden states to compute semantic deviation, which restricts its applicability to black-box models such as GPT-4. Additionally, its fixed Top-K pruning strategy may lack flexibility in handling inputs with varying risk levels.

## 7 Acknowledgements

This study is supported by grants from the National Natural Science Foundation of China (Grant No. U23A20315, No. 62425208, No. U22A2097, No. 62122018, No. 62020106008), Shenzhen Science and Technology Program (No.JCYJ20240813114208012), Fundamental Research Funds for the Central Universities, and Natural Science Foundation of Sichuan Province (Grant No. 2025ZNSFSC1463).

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
